# Crop Production and Agricultural Carbon Emissions: Relationship Diagnosis and Decomposition Analysis

**DOI:** 10.3390/ijerph18158219

**Published:** 2021-08-03

**Authors:** Jianli Sui, Wenqiang Lv

**Affiliations:** School of Business, Jilin University, No. 2699 Qianjin Street, Changchun 130012, China; jlsui@163.com

**Keywords:** crop production, agricultural carbon emissions, EKC, decoupling, LMDI

## Abstract

Modern agriculture contributes significantly to greenhouse gas emissions, and agriculture has become the second biggest source of carbon emissions in China. In this context, it is necessary for China to study the nexus of agricultural economic growth and carbon emissions. Taking Jilin province as an example, this paper applied the environmental Kuznets curve (EKC) hypothesis and a decoupling analysis to examine the relationship between crop production and agricultural carbon emissions during 2000–2018, and it further provided a decomposition analysis of the changes in agricultural carbon emissions using the log mean Divisia index (LMDI) method. The results were as follows: (1) Based on the results of CO_2_ EKC estimation, an N-shaped EKC was found; in particular, the upward trend in agricultural carbon emissions has not changed recently. (2) According to the results of the decoupling analysis, expansive coupling occurred for 9 years, which was followed by weak decoupling for 5 years, and strong decoupling and strong coupling occurred for 2 years each. There was no stable evolutionary path from coupling to decoupling, and this has remained true recently. (3) We used the LMDI method to decompose the driving factors of agricultural carbon emissions into four factors: the agricultural carbon emission intensity effect, structure effect, economic effect, and labor force effect. From a policymaking perspective, we integrated the results of both the EKC and the decoupling analysis and conducted a detailed decomposition analysis, focusing on several key time points. Agricultural economic growth was found to have played a significant role on many occasions in the increase in agricultural carbon emissions, while agricultural carbon emission intensity was important to the decline in agricultural carbon emissions. Specifically, the four factors’ driving direction in the context of agricultural carbon emissions was not stable. We also found that the change in agricultural carbon emissions was affected more by economic policy than by environmental policy. Finally, we put forward policy suggestions for low-carbon agricultural development in Jilin province.

## 1. Introduction

Modern agricultural activities play an important role in greenhouse gas emissions due to their high material input, energy consumption, and pollutant discharge levels [1,2]. Globally, ensuring food security and coping with climate change caused by greenhouse gas emissions are the most common challenges today. In the past two decades, greenhouse gas emissions from agricultural activities have accounted for 10–14% of total global greenhouse gas emissions (in CO_2_ equivalents) [3]. According to the 2019 Intergovernmental Panel on Climate Change (IPCC) special report on climate change and land [4], greenhouse gas emissions derived from agricultural activities reached 108–191 hundred million tons of CO_2_ equivalents during 2007–2016, and they accounted for 21–37% of global greenhouse gas emissions. Being an agriculture-centered country, China’s sown area has decreased from 117 million hectares to 116 million hectares over the last four decades, while the grain yield has increased from 321 million tons to 664 million tons, which is an increase of 107%. Along with the grain yield per unit area increasing, the amount of chemical fertilizer used increased from 12.69 million tons in 1980 to 52.04 million tons in 2019, and agriculture was the second biggest source of carbon emissions [5]. In this regard, it is necessary for China to study the nexus of agricultural economic growth and carbon emissions.

Considering the ever-increasing speed in crop production, the relationship between agricultural activities and environmental degradation has been intensively studied in recent years, focusing on such areas as the environmental stress on the ecosystem caused by highly specialized crop production [6] and excessive fertilizer application due to the increasing numbers of part-time and aging farmers involved in the process of rural transformation [7,8,9,10]. Meanwhile, different methods have been applied in agri-environmental impact assessment. As is well known, the environmental Kuznets curve (EKC) describes an inverted-U relationship between environmental degradation and income [11]. Up to now, a large body of theoretical and empirical literature has dealt with EKC studies [12,13,14,15]. EKC studies have tested most of the primary pollutants and relevant indicators, including SO_2_, NOx, carbon emissions, ecological footprint, etc. [16,17,18], and both emissions per capita and total emissions [19]; furthermore, these studies have been developed from the earliest simple quadratic functions of the levels of income to multivariate and multi-mediation models, intending to indicate underlying factors, such as globalization, trade openness, human capital, etc. [20,21,22,23]. In particular, many studies have tested the existence of the EKC hypothesis by considering CO_2_ emissions with time-series or panel data; some results support the existence of an inverted U-shape [24,25], some support an N-shape [19], some support an inverted N-shape [21], and some support a linear shape without a turning point [26,27].

In scientific discussions on economic growth versus environmental degradation, EKC deals with the concept of decoupling [28,29]. The Organization for Economic Co-operation and Development (OECD) has described the synchronous changes taking place within economic growth and pollution emissions as different degrees of decoupling/coupling states [30], and the idea of decoupling environmental bads from economic goods has garnered policymakers’ attention, with many scholars assessing the progress in the environmental-degradation–economic-growth relationship. In addition to the decoupling index proposed by the OECD [30], Tapio’s elasticity coefficient has commonly been used and extended as a decoupling index that overcomes the shortcoming of selecting a base period [31]. In terms of empirical studies dealing with decoupling analysis in China, Tian et al. [32] used Tapio’s model to analyze the relationship between agricultural activities and carbon emissions at the national level, and we found that weak decoupling and strong decoupling occurred most during the years 2001–2010, while Yang et al. [33] and Chen et al. [34] conducted further in-depth case studies of China’s main grain-producing areas.

In policy terms, decomposition analysis is increasingly used in environmental policymaking, while the log mean Divisia index (LMDI) method is recommended for general use in the study of CO_2_ emission changes as it meets all constraints, such as complete decomposition, consistency in aggregation, and satisfying the factor reversal test [35,36,37,38,39]. In China, many scholars have used the LMDI method to analyze the drivers of carbon emissions related to crop production on a national or regional basis. For instance, Li et al. [40] pointed out that agricultural economic growth in China had a strong driving effect on carbon emissions, while agricultural production efficiency, agricultural structure, and labor force had certain inhibiting effects on carbon emissions during 1993–2008; Li et al. [2] quantified the contributions of factors influencing agricultural carbon emissions in China from 1991 to 2015 and found that the driving factors varied significantly by region. In fact, in terms of empirical findings, Wei et al. [41] indicated that the state of the agricultural economy was the decisive factor in the increase in crop production carbon footprint, while agricultural investment, urbanization, and technological progress were important factors for reducing the crop production carbon footprint in Guangdong province. Zhao et al. [42] confirmed that the agricultural economic level and industrial structure were the main drivers promoting the increase in agricultural carbon emissions, while agricultural production efficiency and agricultural labor force had inhibitory effects on the increase in carbon emissions in Hunan province. As for Heilongjiang province, Chen et al. [34] showed that agricultural output value was the key driving force of agricultural carbon emission increases, while agricultural production efficiency was the primary driving force for agricultural carbon emission reduction.

The Chinese government has pledged to peak carbon emissions by 2030 and achieve carbon neutrality by 2060. In principle, measures to reduce carbon emissions in manufacturing include technological innovation, environmental regulation, and the market mechanism of carbon trading; however, this is more difficult for the agricultural sector due to its multiple carbon sources, strong randomness and dispersed process, and the limited effects of its market mechanisms. Li et al. [2] indicated that northeast China was the largest contributor to agricultural carbon emissions in the country, and economic factors were the main driving forces of the increase in carbon emissions. Jilin province is a major agricultural province, whose corn yield per unit area ranked first in China for many consecutive years, and it has made great contributions to ensuring national food security. Nevertheless, the agri-environment has been deteriorating daily, and a question in the context of this high yield arises as to the relationship between crop production and agricultural carbon emissions, which seriously affects sustainable agricultural development [43,44]. This paper takes Jilin province as the study area and tests the relationship between crop production and agricultural carbon emissions using EKC and decoupling analysis; then, it decomposes the driving factors of agricultural carbon emissions during 2000–2018, and finally, it puts forward suggestions for agricultural carbon emission mitigation.

The structure of this article is organized as follows. Section 2 explains the methodology and data. Section 3 presents the empirical results, using EKC and decoupling analysis to examine the relationship between crop production and agricultural carbon emissions, and then further decomposes the influencing factors of agricultural carbon emissions. Section 4 contains the discussion and policy implications. Section 5 concludes the study.

## 2. Methods and Data 

### 2.1. CO_2_ EKC Model

Based on the analysis framework of EKC proposed by Friedl and Getzner [19], Zhang et al. [45], and Lv et al. [46], considering the simulation accuracy of the CO_2_ EKC model, this paper first describes the general functional form as a cubic polynomial and builds a cubic curve equation on the basis of two indicators: crop production (expressed as agricultural output value) and agricultural carbon emissions. In view of the target of carbon emission reduction in agriculture, the total values of agricultural carbon emissions and agricultural output value are chosen instead of the per capita index. Then, the existence of a cubic curve-fitting equation is examined according to stationarity and cointegration tests; if the cubic term is not statistically significant, it will be eliminated. Subsequently, the form of the quadratic polynomial will be re-estimated, etc. The CO_2_ EKC may be U-shaped, inverted-U-shaped, N-shaped, inverted-N-shaped, or another shape, depending on the coefficients of the explanatory variable, and the CO_2_ EKC equation is quantified as follows:(1)$$C_{t}=\beta_{0}+\beta_{1}G_{t}+\beta_{2}G_{t}^{2}+\beta_{3}G_{t}^{3}+\varepsilon_{t}$$
where $C_{t}$ denotes the agricultural carbon emissions in year *t*; $G_{t}$ represents the agricultural output value in year *t*; $\beta_{i}$ is the coefficient of the explanatory variable, and $\varepsilon_{t}$ illustrates the random error term. According to Ekins [47] and Friedl and Getzner [19], if $\beta_{1}$ > 0, $\beta_{2}$ < 0, and $\beta_{3}$ > 0, an N-shaped relationship between crop production and agricultural carbon emissions can arise; if $\beta_{1}$ > 0, $\beta_{2}$ < 0, and $\beta_{3}$ = 0, the inverted-U EKC hypothesis stands, and so on.

### 2.2. Decoupling Index

In this paper, the decoupling index (DI) is constructed following Tapio’s elasticity coefficient [31], and it is computed using the following expression: (2)$$DI_{t}=\frac{\Delta C}{\Delta G}=\frac{(C_{t}-C_{0})/C_{0}}{(G_{t}-G_{0})/G_{0}}$$
where $DI_{t}$ is defined to explore the relationship between crop production and agricultural carbon emissions at two time points, *t* and *0* represent the last period and the base period, respectively, and the time interval is one year in this study; $C_{t}$ and $C_{0}$ indicate agricultural carbon emissions in the last period and the base period, respectively; ΔC represents the change rate in agricultural carbon emissions from the last period to the base period; $G_{t}$ and $G_{0}$ indicate the agricultural output value in the last period and the base period, respectively; and ΔG represents the change rate in agricultural output value from the last period to the base period.

According to the different decoupling elasticity values, six degrees of decoupling/coupling states were divided, including recessive decoupling, weak decoupling, strong decoupling, expansive coupling, weak coupling, and strong coupling. The specific grading standards and decoupling elasticity values are shown in Table 1.

### 2.3. LMDI Method 

Based on the research of Li et al. [2], Wu and Zhang [48], and Li et al. [22], an extended Kaya identity for agricultural carbon emissions in Jilin province can be expressed as Equation (3):(3)$$C=\frac{C}{G}\times \frac{G}{TG}\times \frac{TG}{AL}\times AL=CI\times SI\times EI\times AL$$
where C represents agricultural emissions; G represents the agricultural output value; TG represents the gross output value of agriculture, forestry, animal husbandry, and fishery; AL represents the agricultural labor force scale; CI is defined as agricultural carbon emission intensity and is represented as the agricultural carbon emissions per unit of agricultural output value. This last is also an agricultural production efficiency factor—a decline in CI means agricultural production efficiency increases in a specific region, which directly or indirectly reveals a change in the relationship between crop production and agricultural carbon emissions; SI refers to the agricultural structure, which reflects the share of the agricultural output value in the gross agricultural output value; and EI reflects the agricultural economic development level, considering the agricultural labor force. Table 2 shows the meanings of all the symbols in Equation (3).

We used the LMDI method to decompose the drivers of agricultural carbon emissions in Jilin province into four factors: agricultural carbon emission intensity effect (Δ*CI*), agricultural structure effect (Δ*SI*), agricultural economic effect (Δ*EI*), and agricultural labor force effect (Δ*AL*). Δ*CI* reflects the change in agricultural carbon emissions intensity, Δ*SI* reflects the change in the share of the agricultural output value out of the gross agricultural output value, Δ*EI* reflects the change in agricultural economic growth caused by the agricultural labor force, and Δ*AL* reflects the change in the scale of the agricultural labor force.

Based on plus decomposition, the total effects of agricultural CO_2_ emissions are expressed as Equation (4):(4)$$\Delta C_{tot}=C_{t}-C_{0}=\Delta CI+\Delta SI+\Delta EI+\Delta AL.$$

Each effect in Equation (4) is expressed as follows:(5)$$\Delta CI=\sum \frac{C_{t}-C_{0}}{\ln C_{t}-\ln C_{0}}\cdot \ln\;(\frac{CI_{t}}{CI_{0}})$$
(6)$$\Delta SI=\sum \frac{C_{t}-C_{0}}{\ln C_{t}-\ln C_{0}}\cdot \ln\;(\frac{SI_{t}}{SI_{0}})$$
(7)$$\Delta EI=\sum \frac{C_{t}-C_{0}}{\ln C_{t}-\ln C_{0}}\cdot \ln\;(\frac{EI_{t}}{EI_{0}})$$
(8)$$\Delta AL=\sum \frac{C_{t}-C_{0}}{\ln C_{t}-\ln C_{0}}\cdot ln(\frac{AL_{t}}{AL_{0}}).$$

### 2.4. Data Sources and Processing

In this study, we used time series data from 2000 to 2018 to model the EKC, build a decoupling index, and analyze influence factors using the LMDI method.

The data for quantifying agricultural carbon emissions and the following decomposition analysis were collected from the Jilin Province Statistical Yearbook (2001–2019) and the China Rural Statistical Yearbook (2001–2019). Both the agricultural output value and the gross agricultural output value were deflated to represent a constant 2000 price.

The formula for calculating agricultural carbon emissions is as follows: (9)$$C=\overset{6}{\underset{i=1}{\sum }}E_{i}\times f_{i}$$
where C denotes agricultural carbon emissions derived from crop production activities; *i* represents the type of agricultural carbon sources, including 6 types of agricultural carbon sources (chemical fertilizers, pesticides, plastic films, agricultural machinery, agricultural plowing, and agricultural irrigation); $E_{i}$ represents the amount of agricultural carbon source *i*; $f_{i}$ is the i-type carbon emission coefficient of agricultural carbon sources (derived from Li et al. [2], Tian et al. [32], and Guo et al. [39]), and C, CH_4_, and NO_2_ are converted into CO_2_ equivalents with reference to the IPCC [49].

Table 3 reports the descriptive statistical results of the main variables in this paper.

## 3. Results

### 3.1. Estimating a CO_2_ EKC

Based on time series data from 2000 to 2018, we estimated the CO_2_ emissions derived from crop production in Jilin province (Figure 1). Agricultural CO_2_ emissions reached 3.09 million tons in the first year (2000); after a transitory upward trend, there was a slight decline in 2003, where the annual decline rate slid to −15.3% owing to the fact that farmers’ willingness to grow food hit a low. The Chinese issued a series of policies of agricultural tax reduction and exemption in 2004, and then, the farmers recovered their confidence in agriculture and increased inputs in crop production, leading to agricultural CO_2_ emissions increasing with a growth rate of 36.2% and reaching a phase peak of 5.91 million tons in 2007. Subsequently, there was a clear break in the increase in agricultural CO_2_ emissions due to the global financial crisis in 2008, which then plunged to a new low (annual decline rate of −31.9%), which was even lower than in 2004. Under the stimulus of beneficial farming policies in 2009, agricultural CO_2_ emissions climbed steadily and reached 6.83 million tons in 2018. The Chinese government put forward a policy of building a resource-conserving and environmentally friendly society in 2012, under its influence, the growth rates of agricultural CO_2_ emissions declined in a fluctuating pattern during 2012–2018, while agricultural CO_2_ emissions still showed a rising trend.

Figure 2 shows a scatter plot of agricultural CO_2_ emissions and agricultural output value at a constant 2000-level price in Jilin province. An N-shaped relationship between crop production and agricultural carbon emissions can be seen. There were three obvious turning points in agricultural CO_2_ emission levels during 2000–2018 in Jilin province, including a slight decline in 2003 (3.13 million tons of agricultural CO_2_ emissions and 425.27 hundred million yuan of value added in agriculture), one phase peak in 2007 (5.91 million tons of agricultural CO_2_ emissions and 516.51 hundred million yuan of value added in agriculture), and then a low in 2008 (4.02 million tons of agricultural CO_2_ emissions and 594.51 hundred million yuan of value added in agriculture). This pattern can be explained by the agricultural policies and macroeconomic factors. Agricultural CO_2_ emissions bottomed out in 2003 and rose after a series of agricultural policies issued in 2004, until being pushed back down again due to the global financial crisis in 2008; they increased sharply thereafter from 2009 to 2018, which was accompanied by the improving agricultural economy. From Figure 2 alone, we cannot infer the future relationship between agricultural CO_2_ emissions and agricultural economic growth in Jilin province, that is, there is no indication of when another turning point will occur in the upcoming period.

Importantly, the scatter plot presents only the correlation between crop production and agricultural CO_2_ emissions during 2000–2018—a causal relationship cannot be identified without undertaking a statistical test, so an additional theoretical analysis is required.

The next analysis framework in this study, following Friedl and Getzner [19] and Zhang et al. [45], was as follows: (1) testing for stationarity of both dependent and independent variables in the time series; (2) examining whether both variables (agricultural CO_2_ emissions and agricultural output value) are cointegrated; (3) further testing the most suitable functional form for depicting the development of agricultural CO_2_ emissions in Jilin province.

Firstly, we conducted a unit root test. In order to overcome the defects of the small samples and prevent sequence spurious regression, we applied the ADF test to examine the stationarity of the dependent variable (CO_2_ emissions, expressed as C) and the independent variable (agricultural output value, expressed as G) for 2000–2018. Table 4 shows the results of the unit root test, showing that both variables chosen in this paper have a significance level of 5% and are stationary series, which can be further tested to determine the long-term equilibrium relationship.

Secondly, as a preliminary step for testing the EKC hypothesis, we conducted a Granger causality test to examine the correlation between the selected variables. Table 5 shows the results of the Granger causality test; the agricultural output value is the Granger cause of agricultural CO_2_ emissions at a significance level of 1% (not vice versa).

Finally, we constructed an empirical EKC model for Jilin’s agricultural CO_2_ emissions. Based on the above test results, the regression model of agricultural CO_2_ emissions and agricultural output value adopted the cubic functional form shown in Formula (10) and Table 6.
(10)$$lnC=-501.44+245.34lnG-38.87{\left(lnG\right)}^{2}+2.05{\left(lnG\right)}^{3}$$

According to the analysis framework of CO2 EKC (in Section 2.1), along with the research of Ekins [47] and Friedl and Getzner [19] and the statistical quality of the estimated CO2 EKC (Table 6), we found that the coefficient of the explanatory variable lnG*,*
$\beta_{1},$ is 245.34 > 0; the coefficient of the explanatory variable ${\left(\mathrm{lnG}\right)}^{2},\beta_{2},$ is –38.87 < 0; and the coefficient of the explanatory variable ${\left(\mathrm{lnG}\right)}^{3}$, $\beta_{3},$ is 2.05 > 0, which validates the cubic functional form. In addition, the statistical quality of the estimation and the scatter plot (Figure 2) confirm one another, and an N-shaped relationship between crop production and agricultural carbon emissions results—that is, increasing agricultural CO2 emissions in the beginning, a decline in agricultural CO2 emissions in the middle, and an upward trend in agricultural CO2 emissions at the end. In this model, no possible turning point toward a decline in agricultural CO2 emissions appears after 2010, which shows the challenges faced by Jilin province to reduce agricultural carbon emissions.

### 3.2. Decoupling Analysis

As regards the criteria for decoupling/coupling degrees (Table 1), the results of the decoupling of agricultural carbon emissions from crop production are shown in Table 7.

There were four types of decoupling/coupling states between crop production and agricultural CO_2_ emissions during 2000–2018: expansive coupling state occurred for 9 years, followed by a weak decoupling state, which occurred for 5 years, and strong decoupling and strong coupling occurred for 2 years each.

Generally, strong decoupling means a positive change rate in agricultural output value, a negative change rate in agricultural CO_2_ emissions, and negative DI. It indicates that the development model of high inputs and high emissions in exchange for rapid agricultural economic growth is gradually shifting to a development model of low inputs and low emissions, and that the pressure on the rural ecological environment has been alleviated. Table 7 shows that the trend of strong decoupling only occurred in 2003 and 2008, wherein agricultural CO_2_ emissions decreased by −15.3% and −31.9%, respectively, while the agricultural output value increased by 5% and 15.2%, respectively. To some extent, special events contributed to the strong decoupling taking place at these two time points. For instance, farmers’ willingness to grow food had been in decline since 1998 and hit a low in 2003, which decreased inputs into agricultural production and agricultural CO_2_ emissions, resulting in the strong decoupling state in 2003; additionally, in 2008, a similar decoupling state appeared in Jilin province, which was affected by the global financial crisis.

Strong coupling indicates the worst situation, with a negative change rate for agricultural output value, a positive change rate for agricultural CO_2_ emissions, and negative DI. In terms of the occurrence of strong coupling in Jilin province from 2000 to 2018, the agricultural output value declined severely due to severe drought and the aftermath of the global financial crisis in 2007 and 2009, respectively, while change rates in agricultural CO_2_ emissions were positive, which gave rise to strong coupling states in these two years.

Weak decoupling indicates a state with positive change rates in both agricultural output value and carbon emissions, and a value of DI ranging from 0 to 1. Table 7 shows that weak decoupling occurred for 5 years, although not consistently in the time series, as it occurred in the years 2001, 2002, 2011, 2014, and 2017. This indicates that agricultural carbon emission growth is somewhat restrained by the execution of various existing policies and measures; however, the absolute carbon emission reduction is smaller in this period than the agricultural economic growth, so agricultural carbon emissions are still rising, and carbon emission reduction measures should be further implemented.

According to the criteria of the expansive coupling state (Table 1), the change rates of both agricultural output value and agricultural carbon emissions in this period are positive, and thus, agricultural carbon emissions rise rapidly. As the most common outcome for Jilin province, the expansive coupling state accounted for 50% of the whole study period. For example, in 2018, the change rate of agricultural carbon emissions was 0.039, while that of agricultural output value was 0.028, and the DI was 1.432, which indicates agricultural economic growth occurred at the cost of accelerated agricultural carbon emissions in 2018.

Figure 3 and Figure 4 show the variation in the values of decoupling elasticity. The DI clearly presents a fluctuating state, along with the different variation characteristics of both agricultural carbon emissions and agricultural output value. Notably, decoupling occurred only at specific time points, i.e., before the implementation of policies to strengthen agriculture and benefit farmers in 2003, or the global financial crisis in 2008. During 2009–2018, an expansive coupling state appeared for 6 years, inlaid and alternating with weak decoupling states.

In light of the above, there was no stable decoupling between crop production and agricultural CO_2_ emissions in Jilin province during 2000–2018, and it is common that carbon emissions increase when the agricultural economic value grows. Expansive coupling has appeared frequently in recent years, which indicates that the target of decoupling agricultural CO_2_ emissions from crop production remains elusive in the coming years.

### 3.3. Results of LMDI Decomposition

According to the definition of CI in this paper, agricultural carbon emission intensity is calculated, and Figure 5 shows changes in agricultural carbon emission intensity in Jilin province during 2000–2018.

Based on the LMDI model and Equations (4)–(8), the decomposition results of carbon emission changes in agriculture in Jilin during 2000–2018 are illustrated in Table 8, and Figure 6 shows the contribution of four factors to agricultural carbon emissions. The total change in agricultural carbon emissions was 3.74 million tons between 2000 and 2018. Generally, the positive driving factors of agricultural carbon emissions included agricultural economic growth and agricultural structure, with contributions of 5.14 million tons and 0.35 million tons, respectively; agricultural economic growth played an especially significant role in the increase in agricultural carbon emissions in Jilin province, with a contribution of 93.56%. These decomposition results are consistent with those of Li et al. [40] and Guo et al. [39]. Agricultural carbon emission intensity and agricultural labor force were negative driving factors of agricultural carbon emissions, with cumulative contributions of –1.18 million tons and –0.58 million tons, respectively; notably, agricultural carbon emission intensity had the most important inhibitory impact on the change in agricultural carbon emissions, with a contribution rate of 66.99%, and agricultural labor force was also a negative factor that cannot be ignored, with a contribution rate of 33.01%.

Figure 7 depicts the differences in the change characteristics of the four factors.

From the perspective of policymaking, we focused on several key time points in our detailed decomposition analysis aimed at reducing agricultural carbon emissions. According to the above results of the CO_2_ EKC model and decoupling analysis, turning points arose in the years 2003 and 2008, with consequential downward trends in agricultural carbon emissions, and strong decoupling states appeared; meanwhile, the only two negative decomposition effects (−56.6 million tons and −188.7 million tons, respectively) appeared. In 2003, except for the agricultural economic effect, which was a positive driving factor of agricultural carbon emissions, agricultural carbon emission intensity, agricultural structure, and agricultural labor force were all negative driving factors, especially agricultural carbon emission intensity, the contribution rate of which was 89.36%, reflecting its significant inhibitory effect.

In 2008, affected by the global financial crisis, on the one hand, agricultural product prices dramatically declined, and farmers’ engagement in agriculture and their inputs into crop production thus greatly decreased. This led to agricultural carbon emission intensity and agricultural labor force both reaching a low, which resulted in agricultural carbon emissions dropping to −258 million tons and −21.8 million tons, respectively, and the contribution rate of agricultural carbon emission intensity reached 92.2%. On the other hand, considering the sharp decrease in agricultural labor force, the agricultural economic effect—namely, the gross agricultural output value divided by agricultural labor force—reached a peak, although it was far less strong than that of agricultural carbon emission intensity.

Another two important points appeared in 2007 and 2009, with a clear peak in 2007 in the N-shaped CO_2_ EKC, and the beginning of the increase in agricultural carbon emissions after the global financial crisis in 2009; these were the only strong coupling states. Further decomposition analysis indicated that both agricultural carbon emission intensity and agricultural labor force acted as positive driving factors of agricultural carbon emissions, while agricultural structure and agricultural economic effect (a decrease in crop production due to drought or financial crisis) acted as inhibitory driving factors. Contrary to the decoupling states, the increases in both the population working in agriculture and the inputs into agricultural production counteracted the inhibitory effects of agricultural structure and the agricultural economic factor, which brought about the increase in agricultural carbon emissions.

Apart from the above four specific time points, most of the study period showed either an upward trend in agricultural carbon emissions, as seen from the N-shaped EKC, or expansive coupling alternating with weak decoupling according to the results of the decoupling analysis, with various decompositions and combinations. In terms of the varied trends after 2012 when industrial overcapacity was exacerbated, the agricultural economic effect still acted as the major driving factor of increase in agricultural carbon emissions; agricultural structure acted mostly as a positive driving factor; the agricultural labor force became an inhibitory factor, reducing agricultural carbon emissions with the promulgation of agricultural policy and market-oriented reforms in the process of rural transformation; the effect of agricultural carbon emission intensity presented no stable inhibitory trend, and it is important for Jilin province to further pursue its key role in agricultural carbon emission reductions.

## 4. Discussion and Policy Implications

(1) From a policy perspective, according to the requirements for a low-carbon economy, when the economy grows, carbon emissions should increase with either a steady or a negative growth trend. However, it is difficult to achieve this goal in the process of transformation in China. What we can do is to guide the transformation to a low-carbon economy in order to achieve a growth rate of carbon emission intensity that is relatively slower than the economic growth rate. Therefore, it is necessary to evaluate the economy–environment relationship with scientific methods and further conduct decomposition analyses of the influencing factors so as to provide policy suggestions.

Both the EKC hypothesis and decoupling analysis describe the dynamic relationship between economic development and environmental pollution, which have both internal connections and obvious differences. The EKC hypothesis expounds the nonlinear relationship of environmental pollution with the level of economic development, while a decoupling analysis reveals whether the relationship between economic development and environmental pressure changes synchronously. The EKC describes the long-term relationship between CO_2_ emissions and economic growth, but it does not capture the short-term change in a particular year or phase. In contrast, a decoupling analysis measures the extent to which CO_2_ emissions decouple from economic growth in the short term, and its indicators provide real-time data but do not identify long-term trends—i.e., the elastic index of decoupling describes the change rates of economic growth and pollution emissions but ignores the absolute changes in both [50]. In light of the work of Vehmas et al. [28], not only the shape of the EKC but also the decoupling results can stem from economic growth, environmental policy, or some other factors. In fact, the agricultural economic development level, industrial structure, rural labor force scale, and crop production inputs (including pesticides, chemical fertilizer, etc.) all have an influence on agricultural carbon emissions, but the EKC hypothesis and decoupling analysis only explain the nonlinear relationship between economic development and environmental pollution, and the effects of economic growth on environmental pollution cannot be expounded by means of the above methods. With this in mind, the latter method is the most important for policymaking.

In view of this, this paper first employed the EKC hypothesis and decoupling analysis to examine the relationship between agricultural carbon emissions and crop production in Jilin province, a major grain-producing area, and then analyzed the influencing factors via LMDI decomposition, so as to provide a scientific basis for subsequent policymaking.

(2) The major findings in this paper indicate some special relationships among the results of the CO_2_ EKC estimation, the decoupling analysis, and the decomposition analysis for Jilin province. Firstly, in terms of time points, strong decoupling coincided with declines in agricultural carbon emissions and the negative total decomposition effects of LMDI in 2003 and 2008 (as shown in Figure 2, Table 7 and Table 8); additionally, strong coupling was partially connected with the peak in the increase in agricultural carbon emissions (such as in 2007, as shown in Figure 2 and Table 7). Secondly, on the whole, the agricultural economic effect played a significant role in the increase in agricultural carbon emissions in Jilin province, while agricultural carbon emission intensity had the most important inhibitory impact on agricultural carbon emissions (Figure 6), similarly to existing conclusions [32,39]. We also found that the positive or negative effects (or driving directions) of the four factors in agricultural carbon emissions were not stable, and macroeconomic changes and the implementation of related policies can be considered to have had an impact on agricultural carbon emissions in Jilin province during 2000–2018. Third, in terms of the policy effect, we found that agricultural carbon emissions were affected more by economic than environmental policy, based on integrating various time points at which economic policies or environmental policies were issued, such as the years 2004, 2008, 2012, and so on (Figure 1 and Figure 7). The empirical results show that agricultural policies or macroeconomic changes affected farmers’ willingness to grow food and increase production inputs, thus having a direct or indirect influence on the change in agricultural carbon emissions. Therefore, policy implications can be inferred from both agricultural and environmental policy.

(3) In terms of policy implications, there is no doubt that Jilin province will pursue agricultural economic growth, given that it is one of China’s main grain-producing areas; however, it is limited by the pattern of the agricultural economic growth. The existing agricultural carbon emission reduction policies have not been effectively implemented in practice, and the efficiency of agricultural carbon emission reduction needs to be improved. On the one hand, the rapid improvement in rural living standards is an indisputable fact, which will have a reducing effect on total agricultural carbon emissions. On the other hand, agricultural carbon emission intensity, agricultural technology progress, and urbanization each play a role in reducing agricultural carbon emissions by developing energy-saving technology, improving agricultural productivity, adjusting the structure of urban and rural regions, etc.; however, at present, China’s agricultural low-carbon technology is in the initial stage, and control over the overall change in agricultural carbon emissions needs to be further improved.

We should actively promote clean energy in rural areas and improve low-carbon agricultural technologies. Clean energy includes both natural and biomass energy. Advanced equipment and technologies producing low-emission clean energy should be actively introduced, and farmers should be encouraged to use clean energy to replace traditional high-emission energy.

We should increase the investment in science and technology and improve the efficiency of agricultural production through agricultural technology. High-yield and efficient agricultural production technology will provide important support for the development of modern agriculture, as well as being a reliable basic means to achieve the goal of low-carbon agriculture. Government departments should provide special funds for the development of low-carbon agricultural technologies and improve the approach to technology research [48].

We should establish a set of low-carbon agricultural ecological compensation technology systems, increase compensation intensity, and encourage farmers to participate actively in fallow and no-till, so as to reduce carbon sources and increase carbon sinks.

## 5. Conclusions 

The following are the conclusions of the paper:(1)Based on the results of the CO_2_ EKC estimation, a long-term N-shaped EKC was found, which reflects that Jilin province is facing a dilemma between agricultural economic growth and agricultural carbon emissions, and the upward trend in agricultural carbon emissions has not changed with the development of the agricultural economy.(2)In the short term, according to the results of the decoupling analysis, weak decoupling, strong decoupling, expansive coupling, and strong coupling occurred in alteration. Among them, expansive coupling occurred for 9 years in total, followed by weak decoupling, which occurred for 5 years, and then strong decoupling and strong coupling occurred for 2 years each. Strong decoupling occurred in 2003 and 2008, which was related to the macroeconomics and policies at these time points; strong coupling appeared due to a severe drought in 2007 and due to the aftermath of the global financial crisis in 2009. There was no stable evolutionary path from coupling to decoupling during the years 2000–2018, which currently remains true.(3)Based on previous research, we used the LMDI method to decompose the driving factors of agricultural carbon emissions in Jilin province into four factors: agricultural carbon emission intensity effect, agricultural structure effect, agricultural economic effect, and agricultural labor force effect. From a policy-making perspective, we integrated the results of both the EKC and the decoupling analysis, and we conducted a detailed decomposition analysis, focusing on several key time points.

Overall, agricultural economic growth played a significant role in the increase in agricultural carbon emissions, while agricultural carbon emission intensity was the main factor behind the decline in agricultural carbon emissions in Jilin province, especially in the years 2003 and 2008, when turning points toward a downward trend in agricultural carbon emissions and strong decoupling states appeared. Another two important time points in the N-shaped CO_2_ EKC included a clear peak that appeared in 2007 and the start of the increase in agricultural carbon emissions after the global financial crisis in 2009; these were the only two strong coupling states. Different from strong decoupling, both agricultural carbon emission intensity and agricultural labor force acted as positive driving factors of agricultural carbon emissions, while agricultural structure and agricultural economic growth acted as inhibitory driving factors of agricultural carbon emissions.

Most of the study period showed an upward trend for agricultural carbon emissions, as seen from the N-shaped EKC, and expansive coupling states alternated with weak decoupling states, especially after 2010, according to the results of the decoupling analysis, with various decompositions and combinations of driving factors. The efforts toward agricultural carbon emission reduction in Jilin province are clearly still ineffective.

## Figures and Tables

**Figure 1 ijerph-18-08219-f001:**
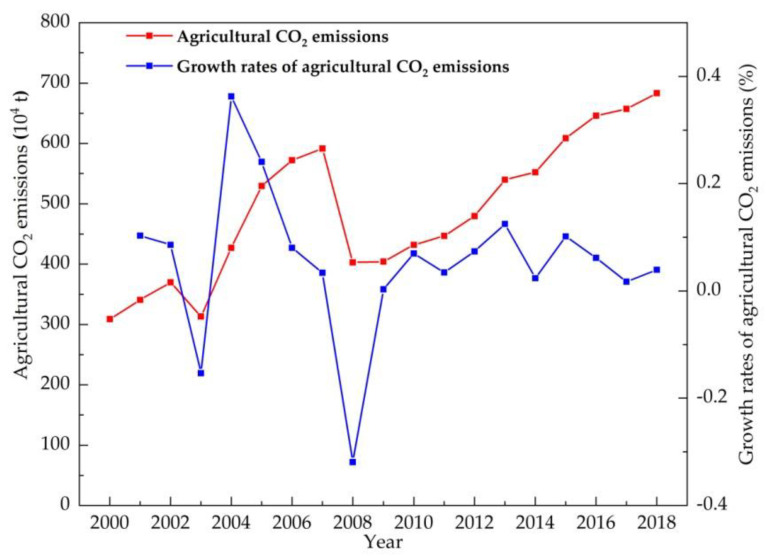
Agricultural CO_2_ emissions and growth rates in Jilin province during 2000–2018.

**Figure 2 ijerph-18-08219-f002:**
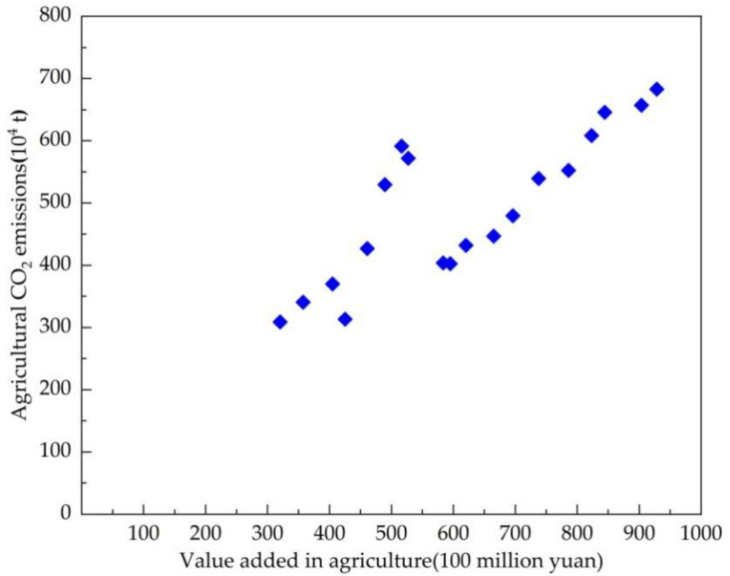
Scatter plot of agricultural CO_2_ emissions and value added in agriculture in Jilin province during 2000–2018.

**Figure 3 ijerph-18-08219-f003:**
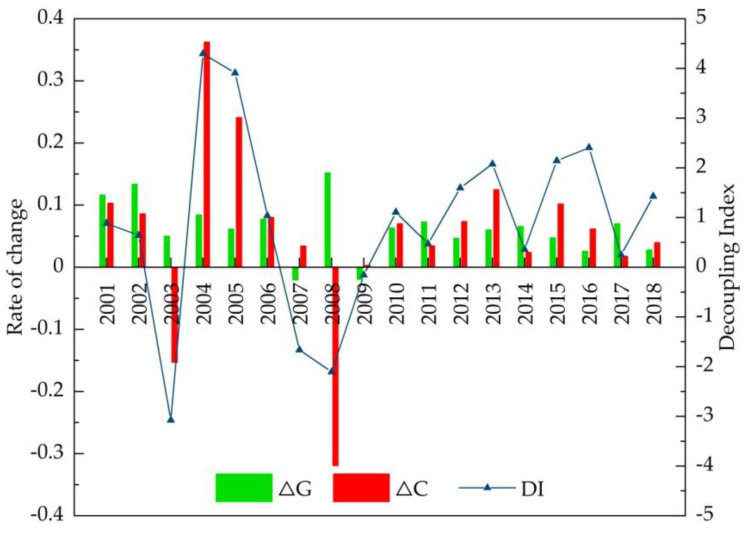
The change in the decoupling index of agricultural carbon emissions.

**Figure 4 ijerph-18-08219-f004:**
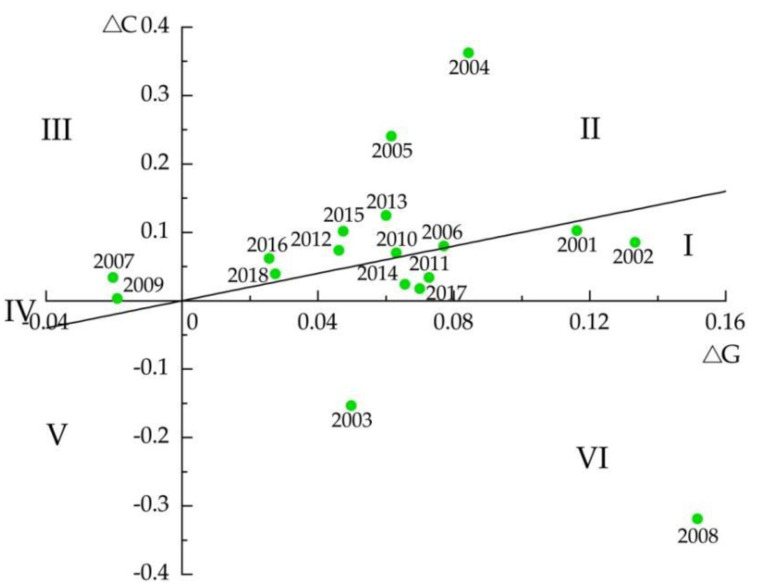
Decoupling distribution of agriculture. Note: I, II, III, IV, V, and VI represent weak decoupling, expansive coupling, strong coupling, weak coupling, recessive decoupling, and strong decoupling, respectively.

**Figure 5 ijerph-18-08219-f005:**
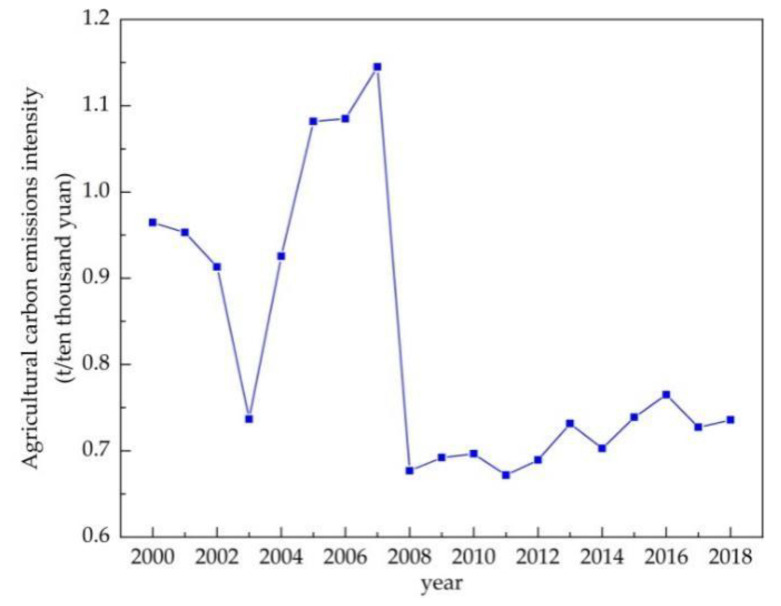
Agricultural carbon emission intensity in Jilin province during 2000–2018.

**Figure 6 ijerph-18-08219-f006:**
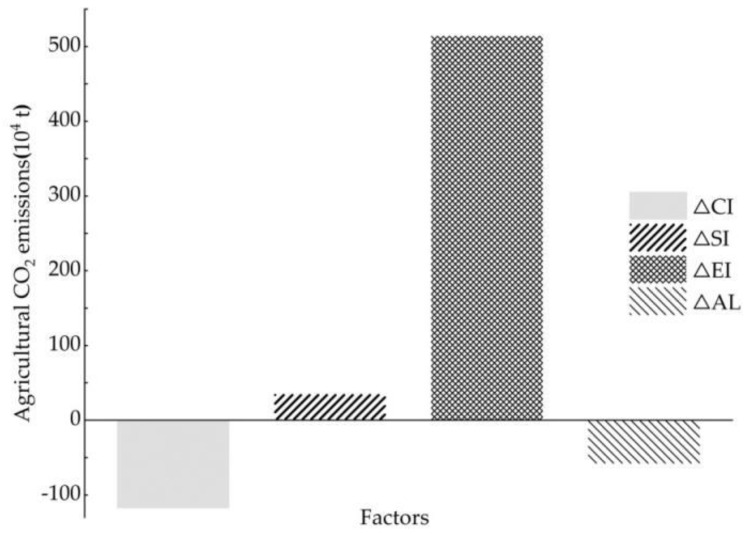
Total contribution of four factors to agricultural CO_2_ emissions in Jilin province.

**Figure 7 ijerph-18-08219-f007:**
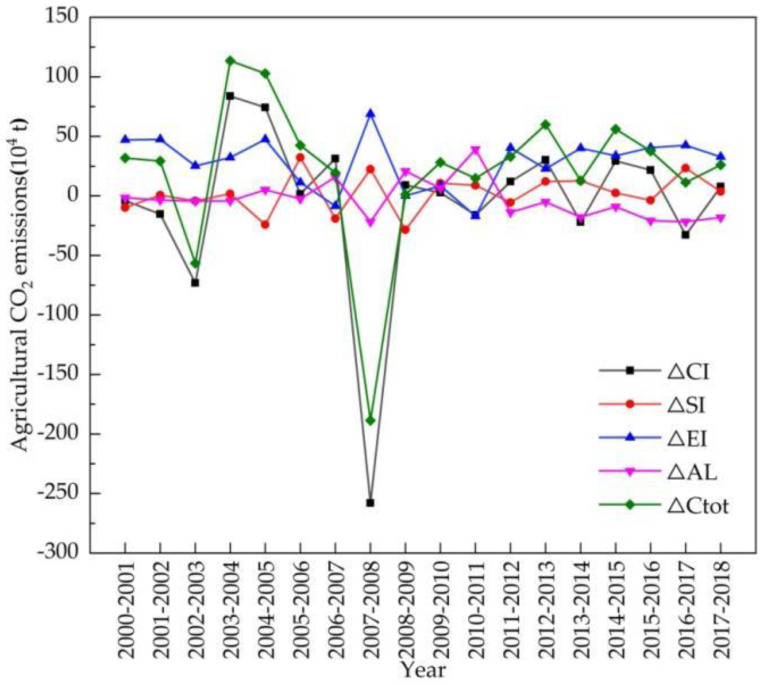
Characteristics of four factors’ contribution to the variation in agricultural CO_2_ emissions in Jilin province during 2000–2018.

**Table 1 ijerph-18-08219-t001:** Degrees of decoupling agricultural carbon emissions from crop production.

Decoupling Degree	Change Rate in Agricultural Carbon Emissions (Δ*C*)	Change Rate in Agricultural Output Value (Δ*G*)	*DI*	Connotation
Expansive coupling	>0	>0	*DI* > 1	Agricultural economic growth occurs at the cost of accelerated agricultural carbon emissions.
Strong coupling	>0	<0	*DI* < 0	The worst state when the agricultural economy is in recession, while agricultural carbon emissions increase.
Weak coupling	<0	<0	1 > *DI* > 0	Agricultural carbon emission reduction rate is slower than agricultural economic recession.
Weak decoupling	>0	>0	1 > *DI* > 0	Agricultural carbon emission growth rate is slower than agricultural economic growth rate.
Strong decoupling	<0	>0	*DI* < 0	The ideal state in which agricultural economy grows, while agricultural carbon emissions decrease.
Recessive decoupling	<0	<0	*DI* > 1	Agricultural carbon emission reduction rate is faster than agricultural economic recession.

**Table 2 ijerph-18-08219-t002:** Definition of variables in the LMDI method.

Variable	Symbol	Definition	Unit
Agricultural carbon emissions	*C*	Carbon emissions derived from crop production	tons
Agricultural output value	*G*	Value added of agriculture	100 million yuan
Gross agricultural output value	*TG*	Total value added of agriculture, forestry, animal husbandry, and fishery	100 million yuan
Agricultural carbon emission intensity	*CI*	Carbon emissions per unit of agricultural output value	tons/yuan
Agricultural structure effect	*SI*	The share of the agricultural output value in the gross agricultural output value	%
Agricultural labor force	*AL*	Rural population engaged in agricultural activities	person
Agricultural economic effect	*EI*	Gross agricultural output value divided by agricultural labor force	yuan per capita

**Table 3 ijerph-18-08219-t003:** Summary statistics of main variables.

Variable	Observations	Mean	Standard Deviation	Minimum	Maximum
Agricultural carbon emissions (10^4^ tons)	19	488.93	119.02	308.96	683.04
Agricultural output value (10^6^ yuan)	19	61,499.98	18,592.87	32,027.00	92,833.79
Gross agriculture output value (10^6^ yuan)	19	12,000.00	33,873.04	60,940.00	17,100.00
Agricultural labor force (10^4^ persons)	19	516.13	24.21	478.24	573.90
Agricultural economic effect (yuan per capita)	19	23,213.92	6665.63	11,791.80	35,696.27
Agricultural carbon emission intensity (tons/10^4^ yuan)	19	0.82	0.16	0.65	1.15
Agricultural structure (%)	19	0.51	0.02	0.48	0.54

**Table 4 ijerph-18-08219-t004:** Unit root test results.

Variable	Test Type	ADF Test	Critical Values at Significance Level			*p*-Value	Test Results
	(c, t, q)	Statistics	1%	5%	10%		
lnC	(c,t,3)	−3.833	−4.728	−3.760	−3.325	0.044	Stationary
lnG	(c,t,0)	−3.879	−4.572	−3.691	−3.287	0.036	Stationary

Note: In the test type (c, t, q), c, t, and q represent the constant, time trend, and lag order, respectively. The lag order is obtained based on the SIC criterion.

**Table 5 ijerph-18-08219-t005:** Granger causality test results.

Null Hypothesis	F Statistics	*p*-Value	Test Results
lnG is not the Granger reason for lnC	14.655	0.003	Reject
lnC is not the Granger reason for lnG	0.835	0.550	Do not reject

**Table 6 ijerph-18-08219-t006:** Estimation of agricultural CO_2_ EKC for Jilin province.

Explanatory Variables	Coefficient of Explanatory Variables
Constant	−501.44 *
lnG	245.34 *
${\left(\mathrm{lnG}\right)}^{2}$	−38.87 *
${\left(\mathrm{lnG}\right)}^{3}$	2.05 *
Adjusted R^2^	0.71
F statistic	60.99

Note: * is at the significance level of 10%.

**Table 7 ijerph-18-08219-t007:** Decoupling states in agricultural carbon emissions in Jilin province during 2000–2018.

Year	Change Rate in Agricultural Carbon Emissions (Δ*C*)	Change Rate in Agricultural Output Value (Δ*G*)	DI	Decoupling States
2000–2001	0.103	0.116	0.886	Weak decoupling
2001–2002	0.086	0.133	0.643	Weak decoupling
2002–2003	−0.153	0.050	−3.075	Strong decoupling
2003–2004	0.362	0.084	4.300	Expansive coupling
2004–2005	0.241	0.062	3.909	Expansive coupling
2005–2006	0.080	0.077	1.040	Expansive coupling
2006–2007	0.034	−0.020	−1.664	Strong coupling
2007–2008	−0.319	0.152	−2.105	Strong decoupling
2008–2009	0.003	−0.019	−0.155	Strong coupling
2009–2010	0.070	0.063	1.109	Expansive coupling
2010–2011	0.034	0.073	0.473	Weak decoupling
2011–2012	0.074	0.046	1.597	Expansive coupling
2012–2013	0.125	0.060	2.078	Expansive coupling
2013–2014	0.024	0.066	0.360	Weak decoupling
2014–2015	0.102	0.047	2.143	Expansive coupling
2015–2016	0.062	0.026	2.408	Expansive coupling
2016–2017	0.017	0.070	0.247	Weak decoupling
2017–2018	0.039	0.028	1.432	Expansive coupling

**Table 8 ijerph-18-08219-t008:** Decomposition results of agriculture carbon emission changes in Jilin province (10^4^ t).

Year	Agricultural Carbon Emission Intensity Effect (ΔCI)	Agricultural Structure Effect (ΔSI)	Agricultural Economic Effect (ΔEI)	Agricultural Labor Force Effect (ΔAL)	Total Effects (ΔC_tot_)
2000–2001	−3.89	−9.81	47.04	−1.57	31.78
2001–2002	−15.22	0.59	47.45	−3.61	29.21
2002–2003	−73.17	−4.26	25.27	−4.45	−56.62
2003–2004	83.81	1.79	32.18	−4.26	113.51
2004–2005	74.25	−24.24	47.55	5.15	102.71
2005–2006	1.58	32.16	11.26	−2.53	42.47
2006–2007	32.00	−19.04	−8.51	15.60	19.35
2007–2008	−258.01	22.39	68.73	−21.82	−188.72
2008–2009	9.00	−28.54	−0.01	20.75	1.20
2009–2010	2.69	10.74	8.36	6.42	28.20
2010−2011	−15.98	8.74	−17.06	39.13	14.83
2011–2012	12.03	−5.59	40.30	−13.84	32.89
2012–2013	30.17	12.08	22.75	−5.14	59.86
2013–2014	−21.97	12.66	39.97	−17.92	12.74
2014–2015	29.23	2.44	33.60	−9.20	56.07
2015–2016	21.65	−3.73	40.45	−20.88	37.50
2016–2017	−32.88	23.33	42.38	−21.65	11.19
2017–2018	7.71	3.68	32.72	−18.19	25.92
2000–2018	−117.72	35.40	514.42	−58.01	374.09

## Data Availability

China Rural Statistical Yearbook (https://data.cnki.net/trade/yearbook/single/n2019120190?z=z009), and Jilin Statistical Yearbook (http://tjj.jl.gov.cn/tjsj/tjnj/2019/ml/indexe.htm).

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
