# Peer review of "Crop Production and Agricultural Carbon Emissions: Relationship Diagnosis and Decomposition Analysis"

_ijerph, 2021, doi:10.3390/ijerph18158219_

Round 1
Reviewer 1 Report
In general, my opinion of the work is quite positive because the introduction gives an appropriate context for the investigation, the methods are scientifically sound and well explained and the results are relevant and discussed in detail supporting the final conclusions.
Specific comments:
1) in the abstract, initialisms must not appear without due explanation (EKC and LMDI).
2) please consider reviewing the sentence in lines 39-41 as it is not clear enough.
3) the various degrees of decoupling/coupling in Table 1 should be more clearly explained with examples, especially the expansive coupling and recessive decoupling.
4) the term “carbon emission intensity” referred to in the abstract and throughout the text should be clarified as it is crucial for the results of the LMDI decomposition.
5) the conclusion in lines 472-474 should be further justified with the use of the results obtained in the work.
6) a large percentage (approximately 60%) of the literature references are more than 5 years old, and many of these are prior to 2011. Please consider updating the literature review.
Reviewer 2 Report
As a reviewer I have the following remarks.
- Think that the abstract should work also as alone – thus the abbreviation should be full spelled in their first use (EKC, LMDI).
- Line 36 – a similar problem in “According to the IPCC special report “, a rule, if the name is used more than one times, define an abbreviation, if only one time don’t introduce an abbreviation. (Line 69).
- Line 140. Can we use more sophisticated regression, say use splines or nonlinear functions?
- Line 150-156. How the base period is defined? Is it any arbitrary year?
- Please add that CNY it is money. Before its first use – that in the text value is represnetd by CNY.
- Line 201, formula (9) – is it i=1 to i=6? Not sure why to 6.
- Figure 2. As CNY is used to represent yuan thus “100 million NCY”.
- Table 5. “Accept or reject null hypothesis”- H0 and Halternative, we can reject H0 but accept Halternative. Please adjust for this statistical issue. I think "do not reject" rater than accept will correct this issue.
- Formula 10. Do we need so many places after the dot “.” Say rather than -501.4398, rounded -501.44.
- I like Figs 3 and 4.
Thank you
Reviewer 3 Report
Table 3, Observations column needs formatting. All figures presented in table 3 should not exceed 2 significant figures. An accuracy is implied that does not exist. Figure 1 is of poor quality, needs to be redrawn on good quality rather than cut and paste. The X-axis does not start from 0, this is bad scientific practice. Figure 2, both axis does not start from 0, this provides a biased presentation of the data. Therefore, the interpretation given in the text referred to both figures needs to be seen from a different prespective.. Also, there is no statical analysis present in the figures. Table 4 also presents numbers implying an accuracy that does not exist.
Line 290 "therefore, we can possibly say that agricultural carbon emissions" the authors do not provide confidence in their results. From lines 295 to 314 a different line spacing is used. Overall, there is a presentation of data that needs further exploration to be fully-fledged into information.
